# Dissimilar Friction Stir Welding of AA2519 and AA5182

**DOI:** 10.3390/ma15248776

**Published:** 2022-12-08

**Authors:** Ivan S. Zuiko, Sergey Malopheyev, Sergey Mironov, Rustam Kaibyshev

**Affiliations:** Laboratory of Mechanical Properties of Nanoscale Materials and Superalloys, Belgorod National Research University, Pobeda 85, 308015 Belgorod, Russia

**Keywords:** friction stir welding, dissimilar welding, aluminium, microstructure, mechanical properties, AA5182, AA2519

## Abstract

In this study, the friction-stir welding (FSW) technique was successfully applied for joining of AA2519 to AA5181 alloy. Microstructure and mechanical properties of dissimilar FSW joints were investigated by optical microscopy, microhardness, and tensile testing. The deformation behaviour of the welded joints was elucidated via the digital image correlation technique. After welding, the ultimate tensile strength of joints was ~300 MPa and ductility was ~16%. The microhardness values observed at the stir zone were higher than those in the base material AA5182. The produced welds demonstrate nearly 100% (based on AA5182) joint efficiency.

## 1. Introduction

Aluminium alloys are classified into heat-treatable alloys, including copper-containing alloys (2xxx series), magnesium- and silicon-containing alloys (6xxx series), and zinc-containing alloys (7xxx series), and non-heat-treatable, including pure aluminium alloys (1xxx series), manganese-containing alloys (3xxx series), silicon-containing alloys (4xxx series), and magnesium-containing alloys (5xxx series) [1].

Given the excellent combination of low density, high strength, and consequently high stiffness, heat-treatable Al-Cu-Mg alloys have found structural application in the aircraft, transport, and military industries. The main mechanism of their strengthening is age-hardening i.e., formation of nano-scale dispersion of secondary precipitates (θ-Al_2_Cu, S-Al_2_CuMg, and so on) during ageing.

The 5xxx series non-heat-treatable alloys, having Mg as the primary alloying addition, possesses high strength, corrosion resistance, formability, and weldability. Usually, they have been strengthened by cold work (i.e., by the strain hardening mechanism). They are widely used in the transport, automotive, shipbuilding, and aviation industry.

Owing to differences in properties (physical and metallurgical) and strengthening philosophy, joining of dissimilar aluminium alloys is a big challenge [2,3]. Applying conventional welding methods (e.g., fusion welding) leads to poor mechanical properties. A detrimental effect is caused by hot cracking, porosity, eutectic melting, evaporation of alloying elements, and so on [2,4,5,6,7,8]. To avoid the above problems, friction stir welding (FSW) could be employed. It is cutting-edge joining technology in which metal is heated and plasticized by the friction between a non-consumable rotating tool, ping, and the workpieces [2,3,9,10,11,12]. The issue that emerges in welding of dissimilar alloys is that each material responds in its own way at higher temperatures with respect to the deformation mechanism. In addition, the extensive deformation at moderate temperatures causes dynamic recrystallization and recovery, resulting in a fine microstructure of the stir region [2,3,9,11]. So, this makes difficult to arrive at common welding parameters that suit both materials. Therefore, each joint configuration requires its own welding regime to achieve high performance.

Because of the nature of the welding process, FSW results in various microstructural zones, i.e., stir zone, thermo-mechanically affected zone, and heat-affected zone (HAZ, the zone between weld and base material) [2,3,7,9,10,12]. As a rule, the microhardness profiles of age-hardening alloys reveal that HAZ demonstrates the lowest values. On the other side, the hardness of the non-heat-treatable alloys was nearly constant across the friction stir welded zone.

Despite that similar FSW are being extensively studied, over recent years, many researchers have turned their attention to the microstructural evolution of dissimilar FSW. Sound joints were obtained between AA2219-T87 and AA5083-H321 [13,14], AA2024-T6 and AA5083-H321 [6], AA2024-T351 and AA5083-H112 [15,16,17], AA2219-T6 and AA5083-H116 [18], AA2024-T351 and AA5083-H111 [19], AA2014-T4 and AA5083-H111 [8], AA2017-T4 and AA5083-H111 [20,21], as well as many others [2,3,7,12].

Both AA5182 and AA2519 are widely utilized for defence [22,23] and cryogenic [4] applications. Therefore, the results on joining of above alloys could be of academic and industrial importance because this allows to set up different properties of the materials in a single component. The present paper aims to report the investigation dealing with the dissimilar joining of two modern aluminium alloys used in the industry, namely AA2519-T840 and AA5182-H32. A careful literature review indicates that there is no attempt to investigate the effect of the FSW regime on the microstructure and mechanical behaviour of the joints of the above dissimilar alloys. The results may be helpful in the processing strategy of aluminium alloys’ joining because dissimilar welding is a core demand of industries to substitute the traditional joining technologies [2,3,7,9,10].

## 2. Materials and Methods

The materials examined in this study were AA5182 and AA2519 alloys produced by the semi-continuous casting method in Joint Research Center, “Technology and Materials”, Belgorod State National Research University. The chemical compositions and treatment conditions are listed in Table 1.

The 3 mm thick treated plates were friction-stir butt welded using an AccuStir 1004 FSW machine (Appendix A). Based on the results of our previous works [24,25,26,27], two welding regimes were selected in order to evaluate the possible influence of the weld heat input. The welding variables are listed in Table 2. The microstructure of alloy after rolling and ageing have been reviewed in detail in [28,29].

The AA5182 and AA2519 sheets were located on the retreating side and advancing side, respectively. The welding tool was fabricated from a tool steel and consisted of a shoulder of 12.5 mm in diameter and a M5 cylindrical probe (pin) of 2.7 mm in length (Appendix A). The welding direction (WD) was parallel to the rolling direction (RD) of both alloys.

The microstructural changes on the post-weld samples were examined using a metallurgical optical microscope Olympus GX71. Specimens for characterization were mechanically ground, polished, and etched with standard Keller’s reagent.

The mechanical properties and deformation behaviour of the produced joints were examined using transverse tensile tests according to the ASTM E8M (Standard Test Methods for Tension Testing of Metallic Materials). Smooth «dog-bone» shape tensile specimens were machined perpendicular to the welding direction (WD). The location of the specimens is shown in Figure 1. The gauge section (35 mm in length and 7 mm in width) of the specimens includes all characteristic FSW zones. The surfaces of tensile specimens were polished to eliminate defects and achieve a uniform thickness. A random ink pattern was applied to the specimens’ surfaces and a high-speed digital camera was used for recording the strain localization phenomena that occur during the tension of joints. Tests to failure were carried out using an Instron 5882 testing machine equipped with a commercial Vic-3D™ system provided by Correlated Solutions, Inc. (Irmo, SC, USA). Micrographs were analysed using digital image correlation software at different stages throughout a tensile test. The initial strain rate was 1.3 × 10^−3^ s^−1^. Vickers microhardness measurements were performed using a Wolpert 402MVD tester(Bühler, Leinfelden-Echterdingen, Germany) using a 2N load and dwell time of 10 s. Samples for mechanical testing and microstructural examination were obtained using a wire electrical discharge machine, according to Figure 1.

## 3. Results and Discussion

### 3.1. Metallography

The visual inspection of dissimilar joints fabricated at both conditions revealed smooth surfaces without any imperfections and macro defects (Figure 2).

Macrographs of the transverse cross section of the produced joints are presented in Figure 3. It is evident from the micrographs that the selected FSW parameters produced defect-free structures. As AA2519 and AA5182 alloys have different etching responses, the metal flow from two sides was clearly visible in the welded joins. Interestingly, because grain refinement took place at the stir zone, the grain boundaries are difficult to distinguish.

### 3.2. Mechanical Testing

#### 3.2.1. Microhardness Survey

Heat generated at the stir zone during FSW softens the material, which promotes the material flow around the tool pin with ease required for efficient stirring of materials. To explore the heterogeneity of the obtained welds, microhardness profiles were measured. The variations in Vickers microhardness along the mid-thickness line of the transverse cross section across the obtained dissimilar welds are presented in Figure 4.

The microhardness of heat-treated AA2519 BM was noticeably higher than that of AA5182 base material. As can be clearly seen, there is a microhardness down trend from unaffected AA2519 to AA5182. The microhardness of the stir zone was on average 20% higher compared with that of AA5182. This is in good agreement with the mixture and grain refinement of the two studied Al alloys sheets in this zone. The gradual reduction of AA2519 microhardness ascribes to precipitation growth and/or dissolution of strengthening precipitates (θ″/θ′ and Ω-phase) owing to the exposure during welding [24,25,26,27]. Transformation from θ′ and Ω to θ leads to a drop in strength/hardness because of the large size of the incoherent phase, huge interparticle spacing (larger than the dislocation slip length), low precipitates’ density, and absence of strength fields around. Similarly, a decrease in microhardness of the AA5182 alloy side can be attributed to the loss in cold working (i.e., annealing effect) during FSW [3]. The authors of [13,18] have reported an analogous hardness reduction in dissimilar welds of AA5083 and AA2219.

#### 3.2.2. Tensile Tests

Hardness and tensile properties are interrelated and both are strictly connected to microstructural changes. In order to better understand the performance, the produced welds were subjected to tensile testing. The obtained curves are presented in Figure 5 and the main numerical results are listed in Table 3. In addition, the typical appearance of the failed tensile tests specimens is shown in Figure 5.

One can easily deduce that the plastic flow of produced joints and AA5182 BM is unsteady. The phenomenon of repeating oscillations in the plastic region of stress–strain curves is a manifestation of the Portevin-Le Chatelier effect [1,30,31]. This follows from alternating pinning and unpinning of moving dislocations during plastic deformation by solute atoms (therefore, the character of plastic flow is serrated). It is interesting to note the yield point phenomenon on BM A5182 and its absence on the tensile curves of joints.

As can be seen from the Table 3, AA2519 base material exhibits significantly higher strength compared with AA5182. The welded specimens demonstrated practically identical properties of the operating parameters used to fabricate the joints. This is contrary to our previous studies [24,25,26,27] of AA2519 similar joints, where the microstructure and mechanical properties strongly depend on the heat input conditions. The dissimilar joints revealed only a slightly lower strength compared with BM AA5182-H321; however, their tensile elongation is reduced by nearly 50%. It is interesting that the joints of previous versions of studied alloys (i.e., AA2519-T87 and AA5083-H321) demonstrated an FSW join efficiency of only 61% [13].

Given the FSW nature of dissimilar aluminium alloys, the material placed on the advancing side dominates in the stir zone [12,13,19]. So, we suggested that, by locating the stronger (AA2519 in our study) materials on the advancing side, one can achieve higher joint efficiencies (including elongation).

It is not surprising that tensile specimen fractured from the weakest point in the softer region, i.e., in the weaker AA5182 (as be will be shown in Figure 4 and Figure 5), away from the weld. This indicates that joints are free of defects and the achieved joint efficiency is around 100% (based on alloy AA5182). This is much higher than what can be achieved with electron beam [5,32,33] and TIG [8] welding processes. The presented data agree with the results obtained in dissimilar FSWed joints of AA2219 and AA5083 [6,13,14,16,17,18].

#### 3.2.3. Material Flow Behaviour Analysis

Figure 6 and Figure 7 display series of images showing the variation in local strain distributions in the tensile direction (ε_yy_) over the gauge length of specimens of dissimilar FSW joints. The images were obtained with digital-image correlation measurements during tensile testing. As expected, the strain distribution is non-homogeneous, which is caused by the heterogeneous microstructure across the dissimilar joints. It is clearly seen that predominant tensile strain increasingly concentrates in the HAZ of the AA5182 side, where the lowest hardness appears (Figure 4). On the other hand, AA2519 experienced almost no plastic strain.

During tensile testing, deformation is initiated at the AS and next concentrated in the heat-affected zone of AA5182 alloy, and the failure is confined to this region. Anyway, a dissimilar weld can be considered as a good weld, when the failure takes place in the weaker of the two dissimilar materials away from the stir zone [3,4].

Further study will be necessary of welding parameters on microstructure and properties of dissimilar joints. However, we assume that AA2519 should be subjected to tempers without pre-deformation (e.g., under T4 or T6 temper).

## 4. Conclusions

Dissimilar joining of AA2519-T840 and AA5182-H32 was successfully performed by friction stir welding. From the presented investigation, the following conclusions can be drawn:Under both welding conditions used in this study, defect-free welds were obtained. There is no macroscopic defects or visible porosity across the weld cross section.Regardless of FSW regime, the dissimilar joints exhibited ultimate tensile strength (~300 MPa) as high as that of the base material, but ductility is reduced by 2 times. Failures occur in softer AA5182. Therefore, a joint efficiency of ~100% is achieved.The variations in microhardness and tensile strength of the dissimilar joints were connected to material flow behaviour, loss of cold work in the HAZ of AA5182, and dissolution and coarsening of precipitates of AA2519.The spectacular efficiency due to AA2519 softening does not reach the strength of AA5182.

## Figures and Tables

**Figure 1 materials-15-08776-f001:**
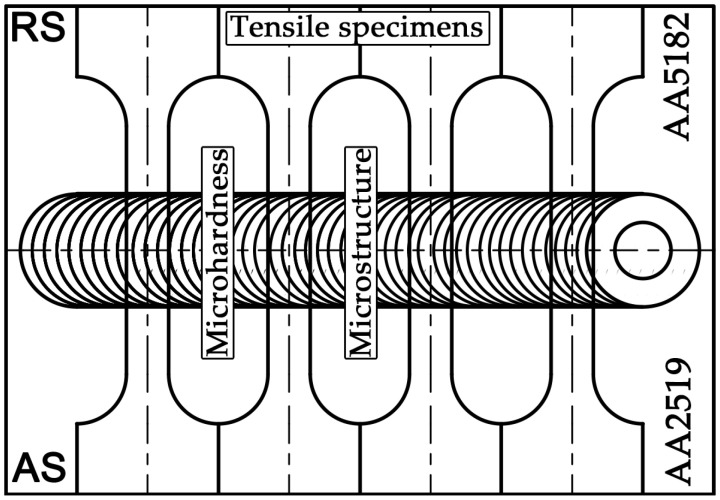
The schematic showing the orientation of tensile test specimens’ relative weld path.

**Figure 2 materials-15-08776-f002:**
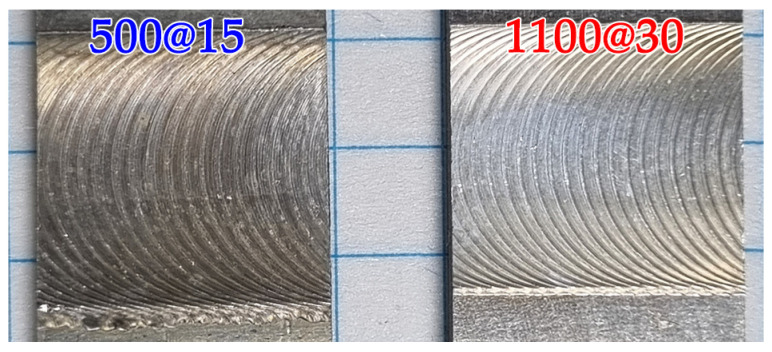
Macroscopic photograph of the dissimilar FSW joints.

**Figure 3 materials-15-08776-f003:**
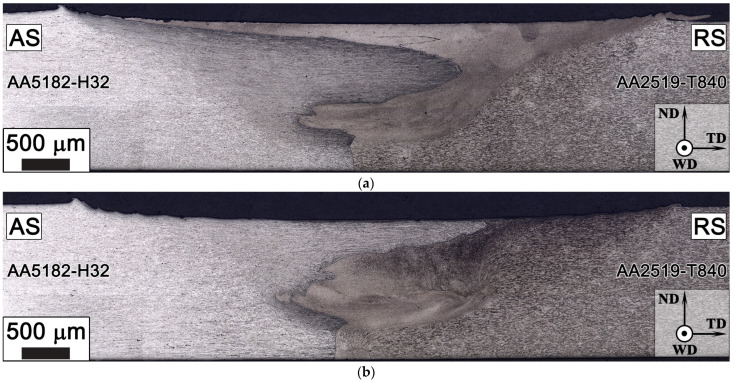
Optical micrographs of 500@15 (**a**) and 1100@30 (**b**) and welds. ND, TD, and WD represent the normal direction, transverse direction, and welding direction, respectively.

**Figure 4 materials-15-08776-f004:**
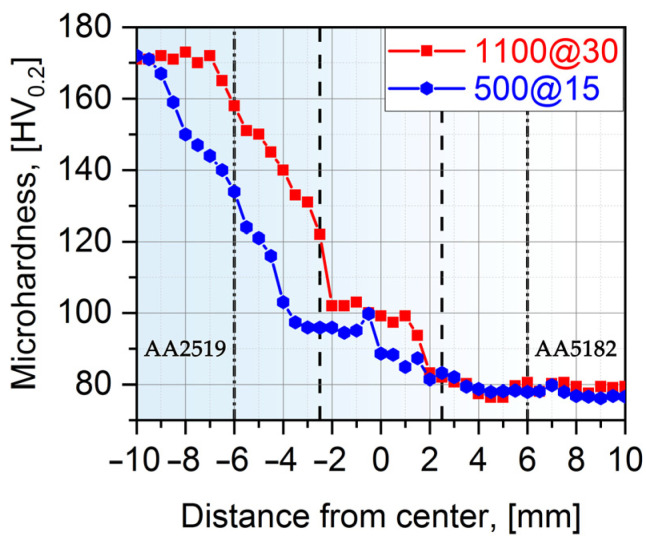
The microhardness profiles across the welds. For the sake of clarity, the probe and shoulder diameters (5 and 12 mm, respectively) were indicated with dash lines.

**Figure 5 materials-15-08776-f005:**
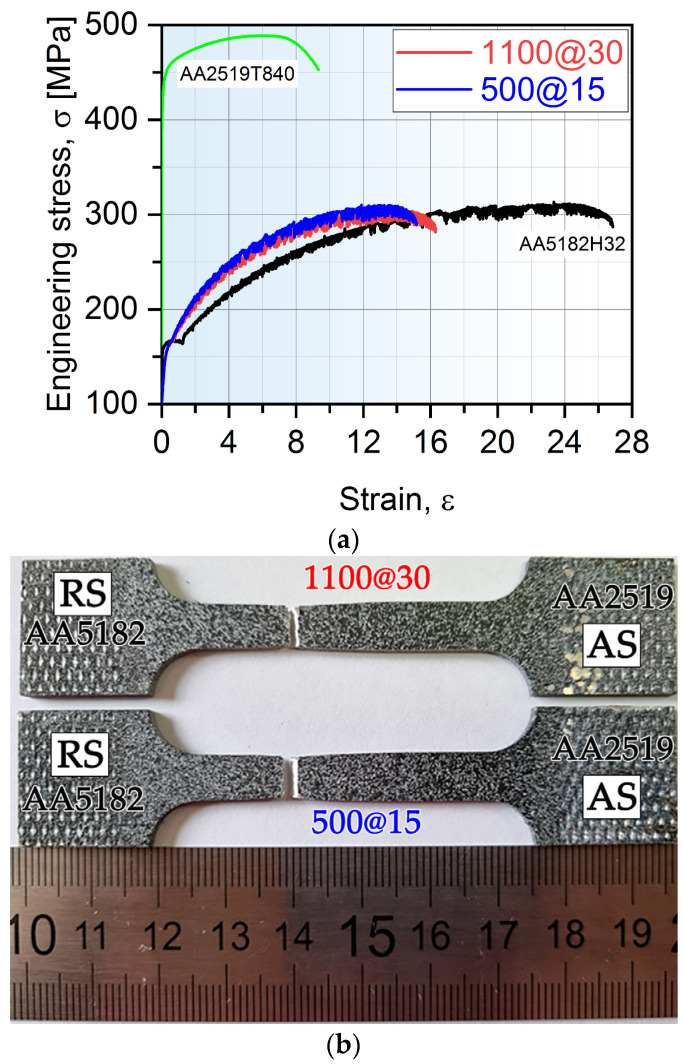
Typical engineering stress–strain curves (**a**) for base materials and dissimilar welding joints; (**b**) the typical appearance of the fractured tensile test specimens.

**Figure 6 materials-15-08776-f006:**
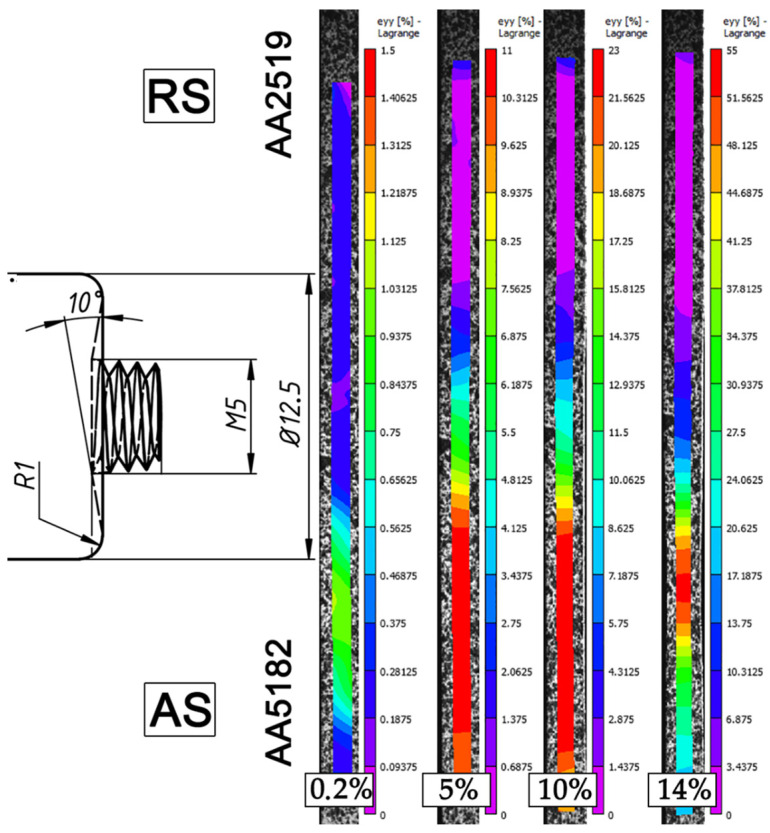
Distribution of local longitudinal strains on the cross-section side of the 500@15 joint, which evolved during transverse tensile tests after total elongation of 0.2%, 5%, 10%, and immediately before failure. For clarity, a tool drawing is added. The tensile direction is vertical.

**Figure 7 materials-15-08776-f007:**
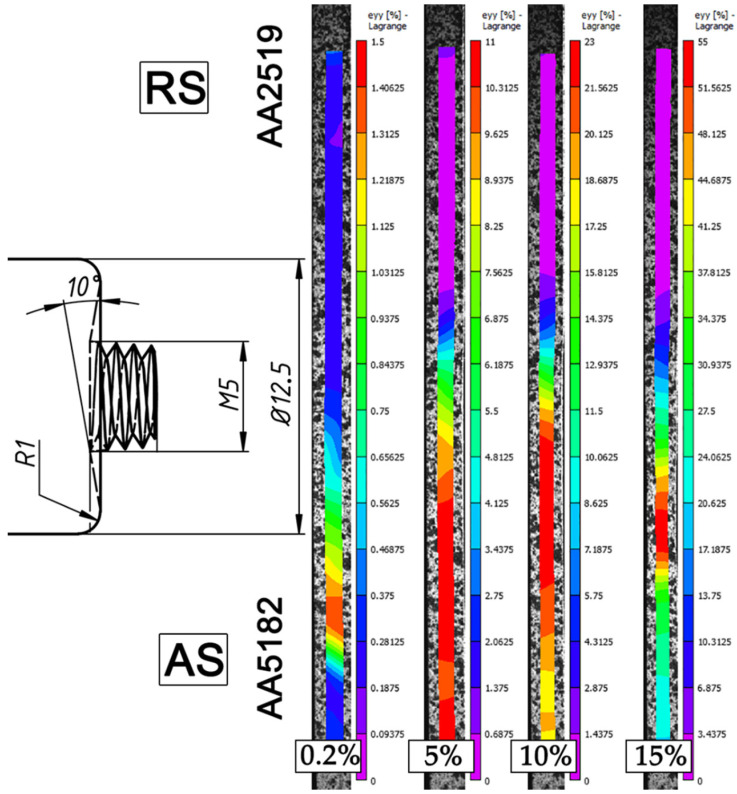
Distribution of local longitudinal strains on the cross-section side of the 1100@30 joint, which evolved during transverse tensile tests after total elongation of 0.2%, 5%, 10%, and immediately before failure. For clarity, a tool drawing is added. The tensile direction is vertical.

**Table 1 materials-15-08776-t001:** Compositions and thermomechanical treatments of the studied alloys.

Alloy	Composition	Primary Processing		Secondary Processing
		Homogenization	Hot Deformation	
AA2519 T840	Al–5.64Cu–0.33Mn–0.23Mg–0.15Zr–0.11Ti	10 h at 380 °C followed by 14 h at 510 °C	rolling at ~425 °C to a strain of 1.4 (75% reduction)	40% cold rolling followed by ageing for 190 °C for a half hour.
AA5182 H32	Al–4.75Mg–0.3Mn–0.15Zr–0.1Ti–0.1Cr	24 h at 360 °C	forging at ~500 °C	75% cold rolling followed by annealing for 300 °C for 1 h

**Table 2 materials-15-08776-t002:** Parameters of the FSW process used in this work.

Weld Designation	Spindle Rate, RPM	Feed Rate, mm/min
500@15	500	380
1100@30	1100	760

**Table 3 materials-15-08776-t003:** Tensile properties of base materials and dissimilar welds.

Alloy/Joint	σ_YS_, MPa	σ_UTS_, MPa	δ, %
BM AA2519	438 ± 2	488 ± 1	8.8 ± 0.9
500@15	162 ± 2	309 ± 1	13.5 ± 1.4
1100@30	162 ± 2	304 ± 1	14.7 ± 1.3
BM AA5182	164 ± 2	313 ± 1	26.8 ± 0.5

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
