# Peer review of "Dissimilar Friction Stir Welding of AA2519 and AA5182"

_materials, 2022, doi:10.3390/ma15248776_

Round 1
Reviewer 1 Report
I have gone through the paper titled “Dissimilar friction stir welding of AA2519 and AA5182”.
Following are my observations.
1. The paper has the latest literature reviews to indicate that the chosen topic has a present scope.
2. Title of table 2 should be changed
3. Reference to table 2 should be given. Or else discuss how range for parameters were chosen.
4. The figure of experimental setup, fixture and tool should be added in paper
5. Better add an image of tensile testing specimen, how its cut from the sample and fracture mode.
6. In line 183 –184 reference missing
7. Line 196, support the result with a macroscopic image of 5x or 10x zoom.
8. An Rt result of weldment can be added to support your conclusion - “There is no visible porosity”.
9. “Joint efficiency of ~100% is achieved" Support with references.
Author Response
The authors would like to express their gratitude to Reviewer and Editor for detailed remarks which allowed improving significantly the quality of the manuscript. Attached file is provided specific replies to the issues raised.

Reviewer 2 Report
The paper is probably publishable, but should be reviersed again in reverised form before it is accepted. I think there are several comments for authors to consider,
1)Please add a figure to illustrate the experimentenal scheme;
2)Please make clear the welding process prameters that were studied in the moment work and its intention for research;
3)It seems the conclusions just told the truth of experiement results and the reason that derived the moment results?
4)Please explain the “SZ”、500@15、1100@30, alike that mention in the article.
Author Response

(The authors gave the same response as above.)

Reviewer 3 Report
The manuscript can be accepted after addressing the following comments.
1) It is inevitable to provide the OM, SEM, EDS redults of different zones of welded material. Due to absence of any image provided, it looks superfacial and speculative to discuss about the precipitation strengtheing, pinning effect etc. So it is must for this manuscript to be published to provide the microstructural details of various zones for both welding conditions.
2) Fig 4, 6 captions should be corrected i.e 500@15 and 1100@30.
Author Response

(The authors gave the same response as above.)

Round 2
Reviewer 3 Report
The manuscript can be accepted in present form.